# Emerging Roles of Exosomes in Stroke Therapy

**DOI:** 10.3390/ijms25126507

**Published:** 2024-06-13

**Authors:** Anthony Larson, Dilmareth E. Natera-Rodriguez, Andrew Crane, Dana Larocca, Walter C. Low, Andrew W. Grande, Jieun Lee

**Affiliations:** 1Department of Neurosurgery, University of Minnesota, Minneapolis, MN 55455, USA; tonylarson3333@gmail.com (A.L.); dilma17n@umn.edu (D.E.N.-R.); atcrane@umn.edu (A.C.); lowwalt@umn.edu (W.C.L.); grande@umn.edu (A.W.G.); 2DC Biotechnology Consulting, Alameda, CA 94501, USA; danaclarocca@gmail.com; 3Stem Cell Institute, University of Minnesota, Minneapolis, MN 55455, USA; 4UniverXome Bioengineering, Inc. (Formerly Known as AgeX Therapeutics Inc.), Alameda, CA 94501, USA

**Keywords:** stroke, exosomes, stem cells, stroke therapy

## Abstract

Stroke is the number one cause of morbidity in the United States and number two cause of death worldwide. There is a critical unmet medical need for more effective treatments of ischemic stroke, and this need is increasing with the shift in demographics to an older population. Recently, several studies have reported the therapeutic potential of stem cell-derived exosomes as new candidates for cell-free treatment in stoke. This review focuses on the use of stem cell-derived exosomes as a potential treatment tool for stroke patients. Therapy using exosomes can have a clear clinical advantage over stem cell transplantation in terms of safety, cost, and convenience, as well as reducing bench-to-bed latency due to fewer regulatory milestones. In this review article, we focus on (1) the therapeutic potential of exosomes in stroke treatment, (2) the optimization process of upstream and downstream production, and (3) preclinical application in a stroke animal model. Finally, we discuss the limitations and challenges faced by exosome therapy in future clinical applications.

## 1. Introduction

Stroke is a leading cause of death for Americans [1]. In the US alone, there are 795,000 strokes each year with an estimated cost of over USD 53 billion [2,3], and the incidence of stroke is increasing as demographics shift to an older population. Substantial racial/ethnic disparities in stroke exist. Stroke mortality rates are higher in Black Americans, American Indians, Alaska Natives, Native Hawaiians, and Other Pacific Islanders, compared with White Americans [4], resulting mainly from unequal medical treatment [5,6]. The COVID-19 pandemic also resulted in increased demand for stroke management drugs as the prevalence of stroke increased due to the spread of SARS-CoV-2 [7,8].

Currently, only one drug, tissue plasminogen activator (tPA), is approved for stroke intervention, and less than 5% of stroke patients ever receive treatment [3], which leaves 760,000 people untreated each year. 

This standard method is often unable to be utilized for a multitude of reasons. For instance, tPA cannot be administered if a patient misses the 8 h treatment window, which often occurs due to a delay in the recognition of stroke symptoms. Transportation limitations and challenges can also be a factor. Others have a contraindication for treatment due to recent surgery, anticoagulation issues, or a recent but separate stroke. The resulting limitations in treatment options lead to increased risk of a brain hemorrhage and neurological deficits. 

Over the last 20 years, the most significant advances in stroke treatment have primarily focused on opening the occluded blood vessel via mechanical thrombectomy. Despite these efforts, clinicians have only been able to expand the treatment window from 6 h to 8 h, shifting the percentage of patients treated from 2.5% to almost 5%, which has allowed an increase in treatment from 20,000 strokes/year to 40,000 strokes/year. However, currently, no drugs are available on the market to promote neurological recovery and neurovascular remodeling [9]. Exosomes address this gap by creating an opportunity to treat the majority of stroke victims. This could lead to a potential paradigm shift in the management of stroke by overcoming the existing current reality in which stroke intervention beyond 12 h is of no benefit.

## 2. Exosomes and Their Potential Role in Therapy

In 2024, minimal information for studies of extracellular vesicles (MISEV) came up with the nomenclature extracellular vesicles (EVs) as a generic term for particles naturally released from cells [10]. EVs are categorized into different subtypes, including exosomes (30–150 nm), microvesicles (MVs, 150 nm–1 μm), and apoptotic bodies (1–5 μm), which are based on their subcellular origin, biogenesis, size, and molecular compositions [11]. Exosomes are originated from the endosomal membrane compartment and stored in intraluminal vesicles within multivesicular bodies of the late endosome. Multivesicular bodies are derived from the early endosome compartment and contain within them smaller vesicular bodies that include exosomes [12]. Exosomes are released from the cell when multivesicular bodies (MVBs) fuse with the plasma membrane. Exosomes can be defined by specific markers of biochemical composition (i.e., CD9, CD81, CD63, Tsg101, and Alix) and physical characteristics such as size, for example, ”small EVs” less than 150 nm in diameter. 

Among the studies included in this review, four described the use of exosomes [13,14,15,16,17], while the others focused on either EVs [18,19,20,21] or MVs [22]. Recognizing that there can be significant variability in the nomenclature applied to exosomes, we further evaluated the processes for which particles were isolated and characterized in stroke studies. In this review, to be fully inclusive of the terminology used across the various studies, we use the term “exosome” to refer to a subtype of EV that ranges in size from 30 to 150 nm as well as specific exosome markers.

Exosomes are secreted by most cell types and play a key role in inter-cellular communication through the transfer of their cargo of lipids, proteins, and RNAs to recipient cells [12]. For example, the transfer of RNA including micro-RNAs (miRNAs) is particularly relevant given their ability to regulate the function and differentiated state of many cell types. Indeed, miRNAs are important controllers of disease-related pathways, including stoke [23]. Moreover, exosomes appear to be important during embryonic development and in regenerative processes following injury, including angiogenesis and vascular repair [24,25,26] and thus have great potential as therapeutic agents for regenerative medicine [27,28]. Exosomes can be isolated from cell media and potentially stored for long-term use because exosome cargo is protected by a lipid bilayer. These properties of exosomes make them a promising therapeutic agent, for they can serve as an efficient pharmacological delivery system to carry miRNAs [29] and siRNAs [30], as well as proteins [31] for targeted therapy. Stem cell-derived exosomes have demonstrated remarkable potential and feasibility of use in a multitude of diseases [32], and several clinical trials are currently underway evaluating the safety and efficacy of exosome therapy for a variety of pathologies including primary cancers, type-1 diabetes, ulcers, and wound healing [32,33,34,35,36].

## 3. Therapeutic Potential for Stroke: Stem Cells versus Exosomes

Currently, 95% of stroke patients are not treated because they were unable to get medical treatment within the narrow window for tPA use and therefore most are left with some permanent neurologic deficit. Recognizing the need for therapies beyond the accepted 8 h time window after stroke, stem cell-based therapies have become attractive alternative treatment for ischemic stroke. Therapeutic cells are derived from various sources including pluripotent stem cells (PSCs), neural stem cells (NSCs), mesenchymal stem cells (MSCs), umbilical cord blood stem cells (UCBSCs), and adipose-derived stem cells (ADSCs) [37,38,39,40]. Multiple studies have evaluated the effect of stem cells in preclinical animal models of ischemic stroke. Recently, several clinical trials using cell transplantation (i.e., MSCs, ADSCs, BMSCs, and UCSCs) have been performed [41,42,43,44] in various diseases. Although there have been promising indications, the efficacy of stem cell therapy in treating stroke has yet to be confirmed in ongoing clinical trials.

The initial concept for stem cell administration following stroke was to regenerate new neurons that integrate into host tissue to replace lost neurons. However, several lines of evidence revealed that systemically grafted MSCs were trapped in the lung, resulting in low cell numbers or even no detectable cells within the ischemic brain [45,46]. This evidence suggests that the positive therapeutic effects of stem cells appear to be largely attributable to paracrine factors via interactions with brain parenchymal cells, which can exert their effects on neuroprotection, neovascular remodeling, and immune modulation. In this regard, it has been reported that treatment with conditioned stem cell media alone has been shown to provide similar benefits as compared to treatment with stem/progenitor cells alone, suggesting that cellular secreted factors are responsible for the treatment effect [47,48]. Treatment using supernatant of cultured MSCs not only enhanced the function of keratinocytes and endothelial cells, but also recruited macrophages to promote the wound healing process [48]. Increasing evidence has indicated that MSCs or ADSCs secrete several growth factors and cytokines, including neurotrophic and neuroprotective factors, angiogenetic factors, and anti-inflammatory cytokines [49,50,51,52,53,54]. 

More recently, exosomes have been isolated from conditioned medium of cultured stem cells, supporting the hypothesis that exosomes secreted from stem cells facilitate cellular communication between stem cells and brain parenchymal cells, leading to a therapeutic effect [20]. The treatment of stroke using exosomes may be preferred over stem/progenitor cell-based therapies because of the inherent challenges with manufacturing and safety concerns of cell therapy. Perhaps of utmost concern with stem cell therapies are their tumorigenic potential [55], the possibility of small vessel occlusion following administration [56], as well as immunogenicity of allogenically derived stem cells [55]. Though transplanted stem cells have been shown to possess intrinsic tropism to the areas of tissue injury, several studies have demonstrated that only a small fraction of the administered cells remain within the injured target tissue [57]. Furthermore, specific and often arduous culture conditions are needed to maintain the viability and function of stem cells, which may pose a challenge for storage and delivery of the cells for immediate use in stroke patients [58]. Recently, reports on exosome-based treatments have shown significant effects in angiogenesis, anti-inflammation, neurogenesis, and anti-apoptosis of stroke [21,59]. In addition to their therapeutic potential, exosomes are likely to be more cost-effective to manufacture, store, and deliver at scale. Early studies suggest they are less likely to induce immune rejection than cell therapies [60]. Thus, exosome therapies have many potential advantages compared to cell therapy for treating stroke.

## 4. Purification Methods and Characterization of Exosomes

Most studies described here used exosomes that were isolated from cell culture conditioned medium. For therapeutic applications, developing a method that can provide intact and pure exosomes is a pivotal step. Several purification methods such as ultracentrifugation [16,20,21,22,61,62,63,64,65,66], ExoQuickTM exosome precipitation [67,68], and miRCURY^TM^ exosome extraction kit [19,20,69] have been applied for stroke research. Ultracentrifugation is the most common method at present. But this method, in which the cell supernatant is first cleared by a low-speed spin, then ultracentrifuged at high speed (>100,000 g) to yield exosomes, can only concentrate substances of similar density and size and thus lacks specificity for exosomes. This results in impurities in the precipitate, including protein aggregates, virion, subcellular organelles, and damaged exosomes, which may cause a reduction in their biological activity [70]. The entire separation process takes over 4 h, and the repeatability is poor and unstable. Even though a density gradient ultracentrifugation method using two or more separation steps with different densities, such as sucrose and iodixando [71], reduces impurity issues, it requires preliminary centrifugation and longer centrifugation time (>16 h), which limits its scalability for clinical application. Various commercial kits have been developed by applying chemical precipitation, immune affinity, size exclusion chromatography, or centrifugation, such as Total Exosome Isolation kit (ThermoFisher Scientific, Waltham, MA, USA), miRCURY Exosome Kit (Qiagen, Hilden, Germany), ExoQuick (System Biosciences, Palo Alto, CA, USA), Exo-spin (Cell guidance systems), etc. However, the purity, quantity, and size distribution of isolated exosomes are significantly diverse [72]. Furthermore, the main limitation for these commercial kits is that they are not suitable for the mass processing of exosomes. 

Therefore, it is crucial to develop standardized methods and quantitative methods for isolating exosomes with rapid, cost-effective, scalable, and reproducible purification techniques for clinical application of exosomes [73]. Recently, several groups have shown that combining methods with ultrafiltration and size exclusion chromatography (SEC) could achieve the production of highly purified exosomes for mass production [74,75]. Ultrafiltration such as tangential flow filtration (TFF) is based on a molecular weight cut off (MWCO) separation method, which is one of the simplest methods to exclude impurities (e.g., protein) while preventing the passage of exosomes [76]. SEC is a separation technology using a column containing porous beads (e.g., Sephadex, Sepharose, Sephacryl, and BioGel P) which separates according to molecular size. In contrast to ultracentrifugation, SEC exosome separation can be accomplished using gravity alone or low-speed centrifugation, making it more likely to preserve the biological function of exosomes [77]. Therefore, combining methods of ultrafiltration and SEC could comprise a simple and economical purification method for handing large-scale exosome production in clinical applications.

Various purification protocols have been published; however, it is difficult to compare the purity of the isolated exosomes, due to the lack of standardized methods to determine the purity of exosomes. Exosome particles have been characterized utilizing visualizing methods such as transmission electron microscopy (TEM) and super-resolution microscopy [78]. The number of particles per volume can be measured by nanoparticle tracking methods such as nanoparticle tracking analysis (NTA), dynamic light scattering (DLS), or Tunable Resistive Pulse Sensing (TRPS) [19,69]. Similarly, researchers characterized particles with surface markers commonly used to identify exosomes, including membrane transport and fusion (Rab, GTPases, flotillin), synthesis of multivesicular bodies (Alix, TSG 101), tetraspanins (CD9, CD63, CD81), and cytoskeleton proteins (heat shock protein, actin, and tubulin). The International Society of Extracellular Vesicles (ISEV) provides the minimal guidelines to fulfill the classification of vesicles as exosomes, such as (1) the quantification of particle number to calculate their ratio to the protein or lipid content, (2) the characterization of exosome proteins including the demonstration of the lack of contaminations originating from particle purification, and (3) the determination of cargo contents and functional assays [79,80,81].

## 5. Exosome Application in Stroke Therapy 

In this review, the results of 22 studies were analyzed for the effect of exosomes in animal models of stroke, including acute ischemic stroke [16,20,51,59,61,63,64,65,66,67], permanent middle cerebral artery occlusion [22,64], subcortical infarct [21,69], hypoxia ischemia [68], and intracerebral hemorrhage [19]. Treatment of stroke with exosomes isolated from stem cells, such as MSCs [20,61,66], ADSCs [22,51,62,63,67,69], BMSCs [65,68], and NPCs [21,59], has clearly demonstrated therapeutic benefits in ameliorating brain injury (Table 1). In Table 1, stem cell-derived exosome therapies in animal models of stroke are described, including the source of exosomes, purification methods, characterization, treatment doses, routes for exosome administration in vivo, and evaluation methods. Table 2 summarizes the therapeutic and mechanistic outcomes of exosome application in a preclinical stroke model. Exosomes are specifically internalized by recipient cells due to their ability to cross the blood–brain barrier (BBB) and enter brain parenchyma [57,68,82,83,84]. It has been reported that the majority of exosomes derived from MSCs reach and accumulate in the infarcted area, in contrast to MSCs themselves, which are mostly trapped within the lung after injection [66], indicating that exosomes are an ideal alternative to stem cells for stroke treatment. With their lack of immunogenicity and tumorigenicity, the capacity to penetrate the BBB, and the potential to be stored and used more feasibly than cell-based therapies, exosomes represent an exciting novel therapeutic avenue for treatment of stroke [82,85,86,87].

### 5.1. Stroke Animal Models

The most common stroke model among the studies included in this review was the middle cerebral artery occlusion (MCAO) injury model (both transient suture and permanent electrocauterization models) in rodent or murine studies. For developing stroke therapy, protection of the brain in acute stroke and enhancement of long-term functional outcomes would be ideal. The Stroke Treatment Academic Industry Roundtable (STAIR) recommends developing therapies which could reduce reperfusion injury and promote neurovascular plasticity and recovery later. Many reports demonstrated promising potential for exosome therapies, which were addressing a neuroprotective and/or regenerative therapeutic outcome. Interestingly, however, there was wide variation in the occlusion times to produce stroke, which included 30 min [20], 50 min [62], and 2 h [61] as well as a permanent occlusion [22]. Such variation in occlusion drastically alters the infarct volume, behavioral deficits, and potentially the secondary processes that result from infarction, such as neuro-inflammation, which subsequently can influence the treatment effect observed in these studies. While this makes it difficult to draw very specific conclusions about the treatment effect of exosomes used to reduce neurologic sequelae after stroke, these studies in general highlight an overall positive effect of exosome therapy for the reduction of neurologic injury after stroke. 

Although these experimental murine and rodent models have yielded valuable insights into stroke, therapies that showed enormous promise in these models have been unsuccessful in clinical translation [88,89,90,91]. An expert panel was assembled by the National Institutes of Health to address these shortcomings [92]. This panel recognized that any single experimental model might not necessarily recapitulate the human pathophysiology of stroke. Because no consensus has been reached regarding the “optimal” model for stroke, both the STAIR and Stem Cell Therapeutics as Emerging Paradigm for Stroke (STEPS) translational research guidelines recommend testing of potential therapies in multiple species and in animals with gyrencephalic brain. To obtain vasculature patterns that are more similar to human subjects, nonhuman primates, dogs, and swine are preferred because the size of the peripheral and intracranial blood vessels enables testing used to treat human stroke [91]. Therefore, translational animal models more reflective of human pathology and improved predictive testing of treatments would be critical for exosome therapy in stroke.

### 5.2. Source of Exosomes in Stroke Therapy

Almost all studies included in this review used exosomes which were extracted from adult stem cells, such as MSCs [20,61,66], ADSCs [22,51,62,63,67,69], and BMSCs [65,68]. However, two studies [21,59] instead used either human embryonic stem cell (hESC)-derived NPCs or human umbilical vein endothelial cells (HUVECs) [93]. This is noteworthy in the broader context of exosome studies investigating their therapeutic potential to treat a wide variety of diseases, where many studies isolate exosomes from other cell sources such as umbilical cord blood endothelial cells, neuronal stem cells, and embryonic stem cells. Recognizing that cellular source impacts the contents of exosomes raises the possibility that exosomes from other cell sources may be more therapeutic for the treatment of stroke than those included in this review. For example, we have found that hESC-derived endothelial progenitors produce more potent angiogenic exosomes than adult MSCs [94]. Additionally, while most studies included in this review used MSCs, there was variability in both the organism and the tissue from which MSCs were isolated, including rats [61,69], mini-pigs [62], and humans [21,62], and from either bone marrow [19,20,61] or adipose tissue [62]. Webb et al. reported that NSC-derived exosomes improved cellular, tissue, and functional outcomes in middle-aged rodents, whereas MSC-derived exosomes were less effective [21]. There are many indications that exosome cargo contents are cell type-specific, therefore affecting the biological properties of the resultant exosomes. Hence, it will be critical to explore several options to find optimal therapeutic exosome sources for the treatment of stroke.

### 5.3. Delivery of Exosomes in Stroke Therapy

Multiple studies included in this review have evaluated the effects of exosomes in animal models of stroke (Table 1). In these studies, exosomes were administered by various delivery routes over a wide range of doses. The route of administration was primarily tail vein (IV, 20 studies), but also included intraperitoneal (IP, one study) [51] and direct injection to lateral ventricle (LV, one study) [95].

The doses and timing for exosome administration were extremely variable. First, some studies [20,21] reported delivering from 2.0 × 10^7^ to 2.7 × 10^11^, while others reported the amount of exosomes delivered as the total in a given volume of conditioned medium [47,48]. The majority of studies reported the dose as the total weight of exosome preparation delivered, which varied from 10 μg [95,96], 50 μg [97], 100 μg [16,19,20,21,79] to 300 μg [62,66] or 100–200 μg per kg rat body [22,63]. However, this method lacks standardization because the number of exosomes delivered depends on the purity (exosome particle number/μg). Most studies administered a single dose of exosomes; however, some reported multiple doses [66] that were delivered at 1 and 4 h after stroke [19,66] and others delivering either at 2, 14, and 38 h or 6, 28, and 48 h after stroke [21]. Given the variability in treatment dosages and timing, it is impossible to compare treatment effect between studies. In general, most exosome studies report pointing out the importance of future studies focused on stroke therapy being consistent. When considering therapeutic treatment of stroke in humans, a single-dose treatment would be ideal, and it will be important for future exosome stroke studies to not only delineate the difference between single-dose or multiple-dose therapies, but also to explore the optimal therapeutic doses.

### 5.4. Biodistribution

Several studies evaluated biodistribution and found exosomes in the brain after intravenous delivery. The earliest time exosomes were detected in the brain was at 1 h after treatment [21], while others found exosomes at 48 h after treatment [69]. Interestingly, while Otero-Ortega et al. [69] found exosomes at 24 h in the brain, Webb et al. [21] found that by 24 h, exosomes were no longer present in the brain, although they were still present in the liver, lungs, and spleen. Chen et al. [62] only looked for the presence of exosomes in the brain at 60 days and did not find any. Unfortunately, we did not identify any study that evaluated the temporal pattern of exosomes trafficking to the brain or clearance. In general, we conclude that exosomes can quickly accumulate in the brain after systemic administration and in some cases may be cleared within days after delivery. It may also be noteworthy that it does not appear that exosome dosage or source appeared to influence trafficking, as the studies listed above used exosomes of varying dosages and from different sources.

### 5.5. Functional Improvement

Most studies demonstrated some degree of functional improvement. However, in general, there was no improvement in functional status at early time points. Otero-Ortega et al. [69] found no improvement at 24 h and 7 days after treatment but did see significant improvement in beam walk, rotarod, and modified Rogers test at 28 days after treatment. Similarly, Webb [21] found no difference in neurological deficit at 48 h after treatment and saw an improvement in only NSC-exosome-treated animals at 96 h after treatment but not in MSC-exosome-treated animals. Several studies demonstrated the improvement in functional outcome between 7 and 28 days after treatment with EVs [20,22,61,62,65,68]. Importantly, in studies comparing exosomes to MSCs or a combination of MSCs and exosomes, there was very little difference observed in the treatment effect between these groups, implying that exosomes can be at least as effective as the cellular source from which they are derived [20,62].

### 5.6. Infarct Volume

Infarct volume was evaluated in studies included in this review, either with MRI or histologically. Several studies found a significant reduction in infarct volume in animals treated after stroke with exosomes [20,21,22,61,62,65,68]. Interestingly, reductions in infarct volume were seen after treatment with exosomes as early as 3 days [62], 4 days [21], and 7 days [22], respectively. The studies by Otero-Ortega et al. [69] failed to show significant reductions in lesion size at early time points (48 h and 7 days) but did show reduction in lesion size at 28 days after treatment with exosomes. Also of interest is that Chen et al. [62] found that while there was a reduction in infarct size with either MSCs, MSCs plus exosomes, or exosome treatment alone, there seemed to be a synergistic effect, with MSCs plus exosomes having the greatest reduction in infarct size [95]. Similar to the functional results, Webb et al. found that the reduction in infarct size was only seen with treatment using NSC-exosomes and not MSC-exosomes [21]. This is notable given that the majority of the other studies showing reduction in infarct size did so with exosomes derived from MSCs. In a slightly different approach, Xiao et al. demonstrated that remote ischemic postconditioning (RIP) prior to MCAO resulted in an increased number of circulating exosomes, which was associated with smaller infarct volumes, as evidenced by TTC staining, suggesting that RIP can induce the production of an endogenous source of exosomes [16]. Intriguingly, exosomes were not found to be present in the brain parenchyma, suggesting a peripheral mechanism which promotes neuroprotection following ischemia–reperfusion injury.

### 5.7. Histological Findings

Several studies demonstrated that exosome treatment following stroke was associated with increased or similar levels of neurogenesis, angiogenesis, oligodendrogenesis, and neurite outgrowth compared to the treatment group with MSCs alone [20,22,61]. Interestingly, these phenomena were observed as early as 1 day post-stroke in one study [22], and at later time points in others [20,61]. Though these data suggest that exosomes can facilitate an early, robust repair response following ischemic stroke, further studies are necessary to elucidate whether exosomes directly interact with endogenous stem/progenitor cells to promote repair or facilitate an endogenous response to injury through an indirect mechanism. In general, exosomes derived from different cell lines tend to have similar neurorestorative and neuroprotective effects. However, through our review, we found that exosomes derived from adipose-derived MSCs [62] and exosomes derived from bone marrow MSCs [19] demonstrated varying neuroprotective capabilities following ischemic injury. These differences in ability to provide neuroprotection and decrease apoptosis following an ischemic injury highlight the idea that exosomes derived from distinct stem cell sources may in fact be loaded with different cargo molecules, in turn resulting in different mechanistic functions, and subsequently varying neurological outcomes following treatment.

### 5.8. Mechanism of Action 

While specific mechanisms of action are still being investigated, the potential therapeutic mechanisms of exosomes appear to include pro-angiogenic, immunomodulatory, neuronal regeneration, and/or neural plasticity regulating processes. Several groups have demonstrated that intravenous administration of MSC-exosomes to an ischemic animal model substantially enhances angiogenesis, anti-inflammatory neuroprotection, and behavior improvement. Reported data show that exosomes not only cross the blood–brain barrier (BBB) [65,98], but also deliver functional cargo, which facilitates angiogenesis and protects against neuroinflammation in stroke [17,59,95,96] (Table 2). Given that the inflammatory response following ischemic stroke can induce harmful neurological sequelae, immunomodulation following ischemic stroke has become an attractive therapeutic option [59,87,99]. Studies reviewed here demonstrate that exosomes can possess robust immunomodulatory functions at differing time points following ischemic stroke, leading to improved functional results [20,21,22,59,64]. It is important to note that the immunomodulatory capacity between studies was not consistent among exosomes isolated from varying cell sources. For example, Webb et al. demonstrated that NSC-derived exosomes were associated with increased circulating anti-inflammatory cells as compared to MSC-derived exosomes [21]. Furthermore, some reports found that exosome treatment was associated with improvement in neuroangiogenesis at 28 days after stroke, but otherwise did not alter the early peripheral immune response [20,66,68]. The therapeutic effects of exosomes are mainly attributed to their powerful ability to transfer molecular cargo (i.e., miRNAs and proteins), which facilitates the reduction in secondary injury and stimulates natural tissue repair mechanisms [17,69,95,96]. These data further emphasize the functional variations between exosomes harvested from different cellular sources. In turn, this warrants further investigation into the chemical and mechanistic characteristics of exosomes harvested from varying sources.

**Table 1 ijms-25-06507-t001:** The list of studies of stem cell-derived exosome application in animal models of stroke.

Studies	Source of Exosomes	Animal Models	Purification Methods	Characterization	Administration Dose	Administration Route	Evaluation Methods and Times for Observation
Xin et al. [61]	Rat bone marrow mesenchymal stem cells (BM-MSCs)	Adult male Wistar rat transient (2 h) MCAO model	Ultracentrifugation	Expression of Alix	100 µg total protein of MSC-derived exosomes	Intravenous (IV) injection at 24 h after stroke	Neurologic severity score (NSS) and foot-fault test at 1, 3, 7, 14, 21 and 28 days
Doeppner et al. [20]	Human BM-MSCs	10-week-old mice transient MCAO	Polyethyleneglycol (PEG) precipitation and ultracentrifugation	Expression of TSG101 and CD81	2 × 10^6^ BMSC-EV	Intravenous injection at 24 h after stroke at 3 consecutive time points (24 h, 3 and 5 days) after stroke	Rotarod, corner test and tightrope test, neurologic severity score (NSS) at 7, 14, and 28 days after stroke
Ophelders et al. [19]	Human MSCs	Ovine model of preterm hypoxia-ischemia (HI) min in sheep fetuses at 102 days of gestation.	Polyethylene glycol (PEG) and low-speed centrifugation	Particle size (99–123 nm) and expression of CD81 and TSG101	2.0 × 10^7^ MSC-EV	Intravenous injection 2 consecutive time points, at 1 h and h4 days post-HI	(1) Baroreceptor reflex on days 0–6 post-HI, (2) Collect seizure burden data continuously until 7 days’ post-HI
Lee et al. [22]	Human adipose MSCs (AD-MSC) exposed to normal rat brain extract (NBE-MSCs), stroke-injured rat brain extract (SBE-MSCs) or not exposed to any extract MSCs	Permanent MCA stroke model in male Sprague-Dawley rats	Ultracentrifugation	N/A	0.2 mg EV/kg rat body weight	the common carotid artery injection 48 h after stroke	Neurologic function (open field, foot fault, beam balance, prehensile traction and torso-twisting) at 0-, 3- and 7-days post-MV injection
Chen et al. [62]	ADSCs and ADSC-exosomes isolated from xenogenic pigs	Mini pigs using the KISOTM System	Ultracentrifugation	Particle size (30–90 nm) using TEM and expression of CD63, TSG101 and ß-catenin	300 µg exosomes	Intravenous injection at 3 h after stroke	(1) Sensorimotor functional (Corner Test) studies on day 0, 1, 3, 7, 14 and 28 after stroke, (2) MRI on days 3 and 28 post-stroke and (3) euthanized 60 days after stroke.
Otero-Ortega et al. [69]	ADMSCs obtained from allogeneic adipose tissue of Sprague-Dawley rats	Ischemic stroke in adult male rats by injection of 1 μL of endothelin-1 or of 0.5 U collagenase type IV into the striatum	miRCURY Exosome Isolation Kit	Particle size (<100 nm) using NanoSight and by the expression of CD81 and Alix	100 µg EV	Intravenous injection at 24 h after stroke	(1) Behavior studies (beam walk, rotarod, modified Rogers test) at 48 h, 7 and 28 days after stroke and (2) MRI imaging performed 7 and 28 days after stroke.
Webb et al. [21]	Human NSCs and human MSCs differentiated from the H9 hESCs	Thromboembolic model of stroke in aged mice.	Ultracentrifugation	Particle size (<300 nm) using NanoSight and expression of CD63 and CD81	three dose regiment of EV with 2.7 × 10^11^ EV	Intravenous injection at 2, 14, and 38 h post-stroke	(1) Cerebral Doppler measurements at 6 and 38 h post-injection (2) novel object recognition (NOR) testing to test Episodic memory
Xiao et al. [16]	Endothelial cells exposed to ischemia (6 h)- reperfusion (24 h) in vitro	Transient remote ischemic preconditioning cerebral I/R (MCAO/R) in parallel to remote ischemic preconditioning (RIP) by temporary clamping of the femoral artery using adult male and female Sprague-Dawley rats	Ultracentrifugation	Particle size (40–100 nm) with a JEOL-1010 TEM and expression of CD63, HSP70 and TSG101 by immunohistochemistry, Western blot and flow cytometry	NA	NA	NA
Han et al. [68]	BMSCs from Wistar rat	Intracerebral hemorrhage (ICH) in adult male Wistar rats	ExoQuick exosome isolation	BCA Protein assay and qNano nanopore-based exosome detection system Alix by Western blot, and electron microscopy	100 μg protein of MSC-derived exosomes	Intravenous injection at 24 h post-ICN	Modified Morris water maze (mMWM), modified Neurological Severity Score (mNSS), and social odor–based novelty recognition tests at days 1, 7, 14, 21 and 25
Huang et al. [63]	(1) rat adipose-derived mesenchymal stem cells (ADSCs) isolated from rat (2) Pigment epithelium-derived factor (PEDF)-overexpressing ADSCs	MCAO model using adult male Sprague-Dawley rats	Ultracentrifugation	Expression of CD9, CD63, CD81, and TSC101	100 μg of EVs per kg	Intravenous injection	Oxygen-glucose deprivation (OGD) experiments
Jiang et al. [64]	miR30d-5p overexpressing rat ADSCs	MCAO model using adult male Sprague-Dawley rats	Ultracentrifugation	Size distribution of ADs-Exos, Nanosizer™ technology (Malvern Instruments, Malvern, UK), transmission electron microscopy (TEM), specific exosome markers CD9, CD63, CD81, and TSC101	80 μg of EVs	Intravenous injection	N/A
Geng et al. [67]	Human MI ADSCs (Age: 57–69 year- old) and miR-126 loaded ADSCs	MCAO model using 8–12 weeks Sprague-Dawley rats	ExoQuick^TM^ Exosome Precipitation Solution	N/A	N/A	Intravenous injection	Foot-fault test and a modified neurologic severity score (mNSS) at days 1, 3, 7, and 14 post-stroke
Liu et al. [65]	Enkephalin overexpressing rat BMSCs	MCAO model using 8–12 weeks Sprague-Dawley rats	Ultracentrifugation	cryo-electron microscopy (cryo-EM) analysis, Nanoparticle tracking analysis, specific exosome markers HSP70, CD63, and TSC101	N/A	Intravenous injection at 12 h post-stroke	NSS test and inclined board test at 1 and 3 weeks
Moon et al. [66]	Rat MSCs (p4) or fibroblasts	MCAO model using Sprague-Dawley rats	Ultracentrifugation	NTA analysis, TEM	10, 30, 100, or 300 μg rMSC-EVs	Intravenous injection at 24 h post-stroke	(1) mNSS test days 1, 7 and 14 after stroke, and (2) The cylinder and ladder rung walking tests at 28 days post-injury
Tian et al. [59]	Neural progenitor cells with RGD-C1C2-fusion	MCAO model using C57BL/6 mice (8 weeks old)	Ultracentrifugation	NTA analysis, TEM	100 μg (2.5–3.7 × 10^10^)	Intravenous injection at 1 h of MCAO and 12 h of reperfusion	N/A
Yang et al. [51]	Hypoxic pre-treated mouse ADSCs	MCAO model using C57BL/6 mice	Ultracentrifugation	TEM and light scattering utilizing Nanosizer (Malvern Instruments, Malvern, UK).	N/A	1 day postoperatively via an intraperitoneal injection.	Sensorimotor functional recovery prior to MCAO and 3, 5, 7, and 10 days post-MCAO was measured Rotarod exam (IITC Life Science, Woodland Hills, CA, USA) to define sensorimotor coordination, the adhesive removal test
Jiang X et al. [96]	Hypoxic preconditioning of neural stem cells (NSCs)	MCAO model using C57BL/6 mice	Ultracentrifugation	BCA protein assay kit, TEM; Nano ZS90 for size and zeta potential; CD9 and CD63 were analyzed via Western Blot; CXCR4 measured using ELISA	10 µg EV	Intravenous injection at 1 Day after MCAO procedure	Complex motor ability on mNSS adhesive removal test, ladder rung task weekly until day 28th
Li et al. [100]	M2 microglia	MCAO model using C57BL/6 mice	ExoQuickTC kit from System Biosciences, Palo Alto, CA, USA.	TEM, Western blot, PKH26 red fluorescent cell linke.	M2-Exos (100 μg/mL) EV	Intravenous injection at 2 h after MCAO	Neuronal apoptosis analysis in the MCAO/R model
Hong et al. [97]	UC-MSCs	MCAO model using 8 weeks male Sprague Dawley rats	Ultracentrifugation	TEM, Western blot for CD63, Alix, and TSG101	50 μg EV	Intravenous injection at 4 h after MCAO once a day per 3 days	Neurologic at 2, 4, and 8 h after the onset of occlusion and then daily until sacrifice.
Wang et al. [101]	Bone Marrow-Derived Mesenchymal Stem Cells (BMSCs)	MCAO male SD Rats	Ultracentrifugation exosome isolation kit from Umibio, Shanghai, China	TEM, Western blot, and NTA	200 µg EV	Intravenous injection at the beginning of reperfusion	Coronal brain section analysis for TTC staining
Zhang et al. [95]	NSCs	MCAO male C57BL/6 mice (age: 7–8 weeks, weight: 22–24 g)	Ultracentrifugation	TEM and NTA (Malvern Nano ZS90)	NSC (5 × 10^5^ NSCs in 5 μL PBS) NSC + Exo (5 × 10^5^ NSCs with 10 μg exosomes in 5 μL PBS)	lateral ventricle injection at 7 days post-MCAO Stereoscopic apparatus (RWD, Shenzhen, China). For (AP + 0, ML-1, DV-2.25 mm)	(1)_TTC staining at 1 and 7 days post-MCAO/R. (2) Measurement of reactive oxygen species (ROS) and inflammation at 3 days post-treatment. (3) Behavioral assessments (balance beam, ladder rung, rotarod, modified neurological severity score) conducted at 0–8 weeks post-treatment. (4) Histological examinations performed at 8 weeks post-treatment. MRI for infarct volume
Xiao et al. [17]	Bone marrow mesenchymal stem cells (BMSCs)	MCAO C57BL/6 mice (male, 8-week)	Total Exosome Isolation reagent from Thermo Fisher Scientific	TEM, Western blot, NTA	100 µg EV	Intravenous injection Once per day for 3 days after MCAO	Neurological evaluations using neurological scoring system.

**Table 2 ijms-25-06507-t002:** Summary of the therapeutic and mechanistic outcomes of exosome application in preclinical stroke model.

Studies	Therapeutic Outcomes	Mechanism of Action
Xin et al. [61]	Enhanced NSS, synaptic plasticity, neurogenesis, and angiogenesis	(1) Bielschowsky silver and Luxol fast blue staining: Increased neurite remodeling and (2) increased synaptophysin immunoreactivity, increased number of BrdU+/Dcx+ cells and BrdU+/vWF+ cells in IBZ *
Doeppner et al. [20]	Improved neurological impairment and brain remodeling, comparing to 1 × 10^6^ MSC, peripheral lymphodemia was reversed no infiltrating monocytes, macrophages, lymphocytes, dendritic cells or neutrophils into the brain	(1) Neuroangiogenesis at 28 days post-stroke: increased NeuN+ cell density, NeuN+/BrdU+ cell number, Dcx+/BrdU+ cell number and CD31/BrdU+ cell number; (2) Reversed peripheral lymphodemia at D6, no infiltrating monocytes, macrophages, lymphocytes, dendritic cells or neutrophils into the brain
Ophelders et al. [19]	Reduced baroreflex sensitivity	(1) Myelin basic protein expression: Reduced white matter injury and (2) IBA-1 immunoreactivity: no impact the normal microglial response to HI
Lee et al. [22]	Reduced infarct volume and improvement in neurologic function was similar in animals treated with either MV isolated from MSC exposed to normal rat brain extract or extract from rat brain after stroke	(1) Increased number of DCX+ cells in the ipsilateral ** SVZ, increased alpha-smooth muscle actin and reduced GFAP+ cells, (2) Increase in the anti-inflammatory cytokines IL-10 and TSG-6 and attenuation of the pro-inflammatory factors TNF-alpha and progranulin
Chen et al. [62]	(1) Sensorimotor function: No difference between treatments of ADSC, ADSC-EV and combination, (2) MRI and histological studies: greatest reduced infarct volumes in the ADMSC plus exosome group, (3) Biodistribution at 60 days: no exosomes or ADMSC	Inflammation, edema, fibrosis, necrosis and apoptosis: greatest reduction in the ADMSC plus exosome group
Otero-Ortega et al. [69]	(1) Significant improvement in the behavioral tests at 28 days and (2) MRI: a decrease in lesion size and improved mean axial diffusivity at 28 days (No difference in functional outcome or MRI at 24 h and 7 days after treatment. (3) Biodistribution: EV found in the brain, lung, liver, and spleen 24 h after administration	Extracellular vesicles (EVs) proteome analysis: hydrolase activity, tubulin binding, protein kinase regulator activity, kinase regulator activity, and catalytic activity promoting white matter repair after stroke
Webb et al. [21]	(1) Significant functional improvements of sensorimotor tests (i.e., balance beam walking, the number of footfalls, hanging wire, and tail suspension performance and declarative memory 14 days post-TEMCAO in aged rodents, (2) reduced infarct volumes (TTC staining), and (3) Biodistribution: the presence of EV in the brain infarct area at 1 h after injection and still present in the liver, lungs, and spleen at 24 h after injection	Circulating M2 macrophages and T regulatory cells analysis: Promoted tissue repair and reduced inflammation by modulating immune responses and facilitated communication between cells in the CNS
Xiao et al. [16]	Reduced infarct volumes using TTC-staining	Reduced the rate of apoptosis through downregulation of Bax and caspase-3 and upregulation of Bcl-2 in SH-SY5Y nerve cells
Han et al. [68]	Significant improvement in the neurological function of spatial learning and motor recovery measured at 26–28 days by mMWM and starting at day 14 by mNSS	Increased newly generated endothelial cells in the hemorrhagic boundary zone, neuroblasts and mature neurons in the subventricular zone, and myelin in the striatum without altering the lesion volume. (1) EBA staining for mature vascular detection, (2) DCX, TUJ1, and MAP2 for neurogenesis and (3) BrdU-positive, indicating that there were newly generated neuroblasts (BrdU-DCX, BrdU-TUJ1) and newly generated immature neurons (BrdU-MAP2) around the hematoma and the SVZ.
Huang et al. [63]	Suppressed MCAO-induced cerebral injury (TTC staining 3 days after MCAO).	Activated autophagy and suppressing neuronal apoptosis.
Jiang et al. [64]	Reduced the cerebral injury area of infarction at day 3 post-stroke	Increased anti-inflammatory cytokines IL-4, IL-10. Suppression of autophagy (Beclin-1 and Atg5) and inflammatory factors, TNF-a, IL-6 and iNOS
Geng et al. [67]	MiR-126 exosomes: significant reduction ischemic stroke and MCAO rats Improved functional recovery	Significant increase of the expression of vWF (an endothelial cell marker) and doublecortin (a neuroblasts marker), suppression of microglial cell by Iba1. Decrease of neuron cell death (TUNEL) and increase of cell proliferation.
Liu et al. [65]	Exosomes crossed the blood-brain barrier Improvement of the neurological score.	Reduction of Neurons Injury: LDH, p53, caspase-3, and NO. Improvement of brain neuron density at days 3 and 7: NeuN.
Moon et al. [66]	Biodistribution: larger amounts of hMSCs were trapped within the lung after injection and rMSC-EVs accumulated in the infarcted hemisphere in a dose-dependent manner (30–300 μg), but not in the lung and liver.	Promoted neurogenesis and angiogenesis: miRNA-184 and miRNA-210 Ki-67 (proliferating cells), DCX (immature progenitor neurons), and vWF (angiogenesis) of both ipsilateral and contralateral hemispheres, 14 days after tMCAO. significantly increased coexpression of Ki-67 and DCX in the subventricular zone (SVZ) of both the contralateral and ipsilateral hemispheres
Tian et al. [59]	Biodistribution: NIRF imaging at 24 h. Accumulation of undecorated EVReN or Scr-EVReN in the liver, followed by the ischemic brain and then the spleens and lungs, whereas the RGD-EVReN had a significantly stronger signal in the ischemic brain	Strong suppression of the inflammatory response (TNFa, IL1b and IL-6). RNA sequencing revealed a set of 7 miRNAs packaged in the EVs inhibited MAPK, an inflammation related pathway.
Yang et al. [51]	Improved cognitive function by decreasing neuronal damage in the hippocampus after cerebral infarction.	Delivery of circ-Rps5, downstream targets, SIRT7 and miR-124-3p, which promoted M2 microglia/macrophage polarization
Jiang X et al. [17]	(1) Survival Improvement: MCAO mice treated with hypoxia-preconditioned exosomes (H-EXO) showed a 25% increase in survival compared to standard exosome treatment. (2) Motor Function Recovery: H-EXO-treated mice exhibited superior motor function recovery, outperforming standard exosome treatment in neurological severity scores and behavioral tests. (3) Sensory Acuity and Motor Ability: H-EXO significantly enhanced sensory acuity recovery, restoration of complex motor abilities, and early recovery in ladder-crossing tests. (4) Infarct Volume Reduction: H-EXO treatment resulted in a substantial reduction in infarct volume relative to the whole brain, as observed through MRI and TTC staining. (5) Pathological Examination: Pathological examination revealed that H-EXO reduced spongy tissue, widened cell gaps, and protected neurons, showcasing improved ischemic brain repair capacity.	miR-216a-5p and miR612: upregulated in hypoxic stem cell-derived exosomes, providing stronger neuroprotection. Hypoxia-inducible factor-1a (HIF-1a): increased MSC-derived exosomes, leading to enhanced vascularization of endothelial cells.
Li et al. [100]	OIP5-AS1 reduced cerebral infarct size, brain edema and mNSS scores in MCAO/R mice. M2 microglia-derived exosomal OIP5-AS1 alleviated neuronal apoptosis in the MCAO/R model.	OIP5-AS1 can alleviate MCAO/R-induced brain damage via the pyroptosis-related proteins indicating that OIP5-AS1 could inhibit the expression of pyroptotic proteins. OIP5-AS1 attenuates neuron damage by reducing the protein stability of TXNIP, thereby inhibiting neuron pyroptosis and reducing CIRI.
Hong et al. [97]	MSC-derived exosomes ameliorated cerebral I/R injury via enhancing circBBS2 expression. circBBS2 served as an endogenous sponger of miR-494 to upregulate SLC7A11, resulting in ferroptosis inhibition.	UC-MSC-derived exosomes protected against H/R-induced ferroptosis in SH-SY5Y cells via delivering circBBS2.
Wang et al. [101]	Infarct volume was decreased more evidently for miR-193b-5p.	miR-193b-5p, which is overexpressed in bone marrow mesenchymal stem cell-derived exosomes. These exosomes mediate the activation of the AIM2 inflammasome and induce cell pyroptosis, a form of programmed cell death. The exosomes are absorbed by OGD/R-induced PC12 cells and ischemic penumbra of cerebral tissue, influencing the inflammatory response and cell death associated with ischemic stroke.
Zhang et al. [95]	Combination therapy (NSCs and exosomes) significantly reduces tissue loss compared to NSC treatment alone. Exosomes further decrease neuronal loss in the postlesional hemisphere. Combination therapy superior therapeutic effects compared to individual treatments. Improved motor function and reduced brain infarction in MCAO/R mice. Accelerated and enhanced therapeutic effects with the addition of NSC-derived exosomes.	Delivery of miRNAs to recipient cells and brain tissues, which then regulate the expression of target genes such as STAT3, PTPN1, and CHUK.
Xiao et al. [17]	BMSC-derived exosomes contribute to functional recovery after ischemic stroke by promoting angiogenesis and reducing neuronal cell damage. BMSC-derived exosome-mediated mitigation of OGD/R-caused cell injury and reduced angiogenesis was dependent on Egr2. Exosomes carrying Egr2 can mitigate brain damage caused by MCAO/R in mice, offering a promising avenue for exosome-based ischemic stroke therapy.	Transferring mRNAs and microRNAs, exosomes with overexpressed microRNA-138-5p from BMSCs can confer neuroprotection to astrocytes after ischemic stroke by inhibiting LCN2. Exosomes derived from BMSCs with overexpressed CXCR4 promote the activation of microvascular endothelial cells during cerebral ischemia/reperfusion injury. Role of Egr2 in this process, which binds to the promoter of SIRT6, enhancing its expression. Increased SIRT6 then suppresses Notch signaling, leading to improved outcomes in cell injury and angiogenesis under OGD/R conditions.

* IBZ: ischemic boundary zone; ** SVZ: subventricular zone; vWF: von Willebrand factor; Modified Neurological Severity Scores (mNSS); Arg-Gly-Asp (RGD).

## 6. Clinical Trials and a Perspective on Potential Future Directions for Exosome Therapy in Stroke

There are currently no FDA-approved exosome products for human application in the United States. In recent years, several universities and research hospitals have performed small-scale Phase I clinical trials using exosomes. In particular, researchers have focused on investigating how to address the challenges associated with their pharmaceutical manufacturing, including scalability, batch-to-batch consistency, adherence to Good Manufacturing Practices (GMP) guidelines, formulation, and storage, along with quality controls, access to the market and relative costs, value for money, and impact on total expenditure.

While there are many examples of therapeutic exosomes at laboratory scale, producing exosomes at industrial scale has remained a major barrier in the development of therapeutic exosomes. The cell source for exosome production should be homogeneous, which is designed to obtain batch-to-batch consistency, However, it is not possible with heterologous primary cells, which vary within a donor and between donors. Therefore, the source of exosomes with well-defined identity and homogeneity, stability, and scalability, all of which allow for optimal production and high potency, will be advantageous for usage of exosomes in future clinical applications. One potential solution is the use of clonally pure hESC-derived progenitor cells as a scalable source of exosomes [94,102]. It is critical to provide cell sources for large-scale production and a manufacturing process developed in Good Manufacturing Practices (cGMP) conditions, which includes in-process testing, quality control release procedures, the standard operating procedure (SOP) for production, and the development of product release criteria for the final exosome product. 

Toxicity and safety should be addressed in stroke preclinical models for the future direction of the clinical application of exosomes. Exosome safety may need to be evaluated with potential acute and long-term toxicities, bioactivity, and durability of observed effects. After administrating exosomes, animals need to be monitored for signs of adverse events such as stroke, respiratory distress, seizures, and renal failure, so that suitable preclinical evaluation can occur before moving into the clinic.

## 7. Conclusions

Exosomes offer an exciting therapeutic option for the treatment of stroke (Figure 1). Given their ease of isolation, potential for “off the shelf” storage, low immunogenicity, and lack of tumorigenicity, exosomes may be a viable alternative to cell-based therapies. To fully utilize the potential of exosomes, a standardized methodology for optimal exosome purification and characterization needs to be established. Depending on parent cell source, exosomes can contain a variety of molecules within their cargo, thereby enabling their diverse protective and restorative functions in the treatment of stroke. Future studies may continue to elucidate optimal methods by which exosomes can be engineered in order to provide the best possible neurological outcomes following stroke.

## Figures and Tables

**Figure 1 ijms-25-06507-f001:**
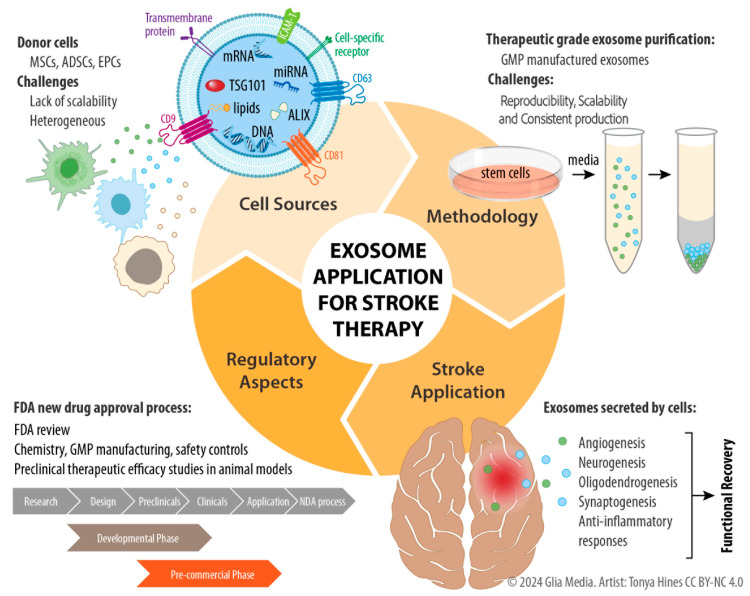
Schematics for the exosome application in stroke therapy.

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
