# Peer review of "Emerging Roles of Exosomes in Stroke Therapy"

_ijms, 2024, doi:10.3390/ijms25126507_

Round 1
Reviewer 1 Report
Comments and Suggestions for Authors
In this review article, the authors present the current state of knowledge on the therapeutic potential of exosomes for use in stroke treatment, with particular attention to any limitations and challenges facing exosome therapy in future clinical applications. The topic of the manuscript is current and potentially of interest to many readers. The review structure needs to be improved. As a potential reader, I am missing information on what exactly exosomes are to be used for in stroke treatment, as drug delivery systems, other targets, what is to be delivered through them? Cargo of exosomes is crucial. The other descriptions are already technical issues, related to, for example, their delivery technique, biodistribution , etc., and the therapeutic benefits obtained. More details on the use of exosomes are in Table 1, but the text of the manuscript is written in very general terms. No synthetic summary.
Additional comments:
1. Line 56: I suggest considering replacing MISEV2018 (reference # 10) with newer recommendations (MISEV2023, PMID: 38326288).
2. Line 57: should be microvesicles (MVs, 150 nm - 1µm), and apoptotic bodies (1-5µm) - lost micro symbol (µ).
3. Lines 107-107: A repetition of the information provided in Chapter 2.
4. Lines 104-122: From this paragraph, all general information on exosomes should be moved to Chapter 2. Left here is only the information that relates to the possibility of using this subpopulation of EVs for therapeutic potential for stroke , according to the title of Chapter 3.
5. Lines 75-155: In my opinion, Chapters 3 and 4 should be combined, since they address the same problem - the limitations of using stem cells in stroke therapy and the benefits that will be achieved by using exosomes for this purpose.
6. Lines 156-210: It's true that there are big problems with the standardization of the exosome isolation protocol. But the fundamental question unanswered in this chapter is: Why isolate exosomes from the conditioned medium (from under which cells) and how will this contribute to stroke treatment? Will exosomes be used as drug delivery systems or for other purposes?
7. Line 272: should be: 2.0x107 to 2.7x1011, lack of superscript
Author Response
We thank the editor and all reviewers for recognizing the significance of our review and for their critical reading and suggestions. In the revised manuscript, we have addressed all the key concerns raised by the reviewers, which has resulted in important revisions to the overall structure of the paper. As a result of these changes, the manuscript has been significantly improved, and we believe it is now suitable for publication in the International Journal of Molecular Science.
Reviewer 1:
“In this review article, the authors present the current state of knowledge on the therapeutic potential of exosomes for use in stroke treatment, with particular attention to any limitations and challenges facing exosome therapy in future clinical applications.”
Response: We thank the reviewer for noting the significance of our submitted review article that has been designed to primarily summarize the therapeutic potential of exosomes derived from stem cells in pre-clinical therapeutic efficacy, safety, and toxicity in the treatment of stroke.
To help you in your review of the revised manuscript, we list below the detailed point-by-point responses are as follows:
- “The topic of the manuscript is current and potentially of interest to many readers. The review structure needs to be improved. As a potential reader, I am missing information on what exactly exosomes are to be used for in stroke treatment, as drug delivery systems, other targets, what is to be delivered through them?”
Response: We thank the reviewer for appreciating the potential interest of this paper to the exosome research community. In the revised manuscript, we have addressed all the key concerns raised by the reviewer, which has clarified the overall review structure and focus.
We certainly agree that exosomes are a promising therapeutic agent, which can serve as an efficient pharmacological delivery system to carry miRNAs and siRNAs, as well as proteins for targeted therapy. In this review paper, however, we intended to focus on therapeutic potential of stem cell-derived exosomes by providing significant information specifically relevant for treating stroke. As we hope you will see, reviewing the other aspect of exosomes as a delivery system is beyond the scope of the current manuscript. That said, and given that we recognize the importance of exosomes as drug delivery systems, we could consider a producing a separate review paper focusing on the role of exosomes as a gene delivery tool and describing their targets for stroke therapy.
- “..The other descriptions are already technical issues, related to, for example, their delivery technique, biodistribution , etc., and the therapeutic benefits obtained. More details on the use of exosomes are in Table 1, but the text of the manuscript is written in very general terms. No synthetic summary.”
Response: In Tables 1 and 2, we have analyzed 22 studies of stem cell-derived exosome therapy in animal models of stroke. Table 1 includes the source of exosomes, their purification methods, characterization, treatment doses, routes for administration in vivo, and evaluation methods. Table 2 summarizes the therapeutic and mechanistic outcomes of exosome application, including biodistribution studies in preclinical stroke model.
- “Line 56: I suggest considering replacing MISEV2018 (reference # 10) with newer recommendations (MISEV2023, PMID: 38326288).”
Response: Thank you. As suggested, we have replaced the reference with the newer MISEV 2023 recommendation.
- “Line 57: should be microvesicles (MVs, 150 nm - 1mµm), and apoptotic bodies (1-5mµm) - lost micro symbol (µ).”
Response: Yes, thank you. We have fixed those discrepancies.
- “Lines 107-107: A repetition of the information provided in Chapter 2. Lines 104-122: From this paragraph, all general information on exosomes should be moved to Chapter 2. Left here is only the information that relates to the possibility of using this subpopulation of EVs for therapeutic potential for stroke, according to the title of Chapter 3.”
Response: Thank you. As the reviewer suggested, we have significantly revised Chapter 2 in order to clearly focus on explaining general information on exosomes and their potential role in therapy, while also including brief information and references on their role as a delivery system.
- “Lines 75-155: In my opinion, Chapters 3 and 4 should be combined, since they address the same problem - the limitations of using stem cells in stroke therapy and the benefits that will be achieved by using exosomes for this purpose.
Response: As the reviewer suggested, we have merged Chapters 3 and 4. Thus, Chapter 3 now addresses the limitations of using stem cells and the benefits exosome in stroke therapy, which have many potential advantages compared to cell therapy for treating stroke.
- “Lines 156-210: It's true that there are big problems with the standardization of the exosome isolation protocol. But the fundamental question unanswered in this chapter is: Why isolate exosomes from the conditioned medium (from under which cells) and how will this contribute to stroke treatment? Will exosomes be used as drug delivery systems or for other purposes?”
Response: In Chapter 5 on exosome application in stroke therapy, we have intensively analyzed the effect of exosomes from 22 studies for stroke treatment. Cellular source impacts the contents of exosomes, so it raises the possibility that exosomes from other cell sources may support more effective therapies for the treatment of stroke. The description was included in this review in the section of Chapter 5.2. Source of Exosomes in Stroke Therapy. The fundamental reason of applying exosomes from stem cells is described in the section of Chapter 3. Therapeutic Potential for Stroke: Stem Cells versus Exosomes. If we recognize the need for therapies beyond the accepted 8-hour time window after stroke, stem cell-based therapies become attractive alternative treatment options for ischemic stroke. Importantly, exosomes isolated from conditioned medium of cultured stem cells were able to suppor the hypothesis that exosomes secreted from stem cells facilitate cellular communication between stem cells and brain parenchymal cells, leading to a therapeutic effect.
- “Line 272: should be: 2.0x107to 2.7x1011, lack of superscript”
Response: Thank you. This has been fixed.
Reviewer 2 Report
Comments and Suggestions for Authors
The review paper entitled Emerging Roles of Exosomes in Stroke Therapy is a well- organized, easy to follow manuscript suitable for publication even in the present form. I recognized only one word-repetition misprint in row 363.
Indeed, the treatment of stroke is almost an everyone concerning illness and any prevention or reducing modality is extremely important. The authors focus on MSCs-derived exosomes treatments, summarizing the possible treatments, the isolation and characterization of exosomes. The review is based on 22 relevant publication, critically presented and well-organized. Some perspectives and problems to be solved are also discussed.
I suggest publication after a careful reading to roll out eventual misprints.
Author Response
We thank the editor and all reviewers for recognizing the significance of our review and for their critical reading and suggestions. In the revised manuscript, we have addressed all the key concerns raised by the reviewers, which has resulted in important revisions to the overall structure of the paper. As a result of these changes, the manuscript has been significantly improved, and we believe it is now suitable for publication in the International Journal of Molecular Science.
Reviewer 2:
- “The review paper entitled Emerging Roles of Exosomes in Stroke Therapy is a well- organized, easy to follow manuscript suitable for publication even in the present form.”
“Indeed, the treatment of stroke is almost an everyone concerning illness and any prevention or reducing modality is extremely important. The authors focus on MSCs-derived exosomes treatments, summarizing the possible treatments, the isolation and characterization of exosomes. The review is based on 22 relevant publication, critically presented and well-organized. Some perspectives and problems to be solved are also discussed.”
Response: We thank the editor and all reviewers for recognizing the significance of our review and for their critical reading and suggestions.
- “I recognized only one word-repetition misprint in row 363.”
Response: Thank you. This has been fixed.
Round 2
Reviewer 1 Report
Comments and Suggestions for Authors
The authors responded to all my comments mentioned in my review and brought to the manuscript the necessary changes that would definitely enhance its quality.